# No-Tillage with Residue Retention and Foliar Sulphur Nutrition Enhances Productivity, Mineral Biofortification and Crude Protein in Rainfed Pearl Millet under Typic Haplustepts: Elucidating the Responses Imposed on an Eight-Year Long-Term Experiment

**DOI:** 10.3390/plants11070943

**Published:** 2022-03-30

**Authors:** Ram Swaroop Bana, Kuldeep Singh Rana, Raj Singh, Samarth Godara, Minakshi Grover, Achchhelal Yadav, Anil Kumar Choudhary, Teekam Singh, Mukesh Choudahary, Ruchi Bansal, Nirupma Singh, Vipin Mishra, Amresh Choudhary, Akshay Kumar Yogi

**Affiliations:** 1ICAR—Indian Agricultural Research Institute, New Delhi 110 012, India; ankit.tiwari2601@gmail.com (A.); ksrana04@yahoo.co.in (K.S.R.); rajsingh221996@gmail.com (R.S.); minigt3@yahoo.co.in (M.G.); achchheyadav@yahoo.com (A.Y.); teekam.singh@icar.gov.in (T.S.); selmukesh@gmail.com (M.C.); nirupmasingh@rediffmail.com (N.S.); amu8805@gmail.com (A.C.); akyogi37@gmail.com (A.K.Y.); 2ICAR—Indian Agricultural Statistics Research Institute, New Delhi 110 012, India; samarth.godara@icar.gov.in; 3ICAR—Central Potato Research Institute, Shimla 171 001, India; 4ICAR—Indian Grassland and Fodder Research Institute, Jhansi 284 003, India; 5ICAR—National Bureau of Plant Genetic Resources, New Delhi 110 012, India; ruchi.bansal@icar.gov.in; 6Yara Fertilizers India Pvt. Ltd., Gurugram 122 010, India; vipmish@gmail.com

**Keywords:** ammonium thiosulphate, conservation agriculture, foliar fertilization, no-tillage, protein, sulphur economy

## Abstract

Yield limitation and widespread sulphur (S) deficiency in pearl-millet-nurturing dryland soils has emerged as a serious threat to crop productivity and quality. Among diverse pathways to tackle moisture and nutrient stress in rainfed ecologies, conservation agriculture (CA) and foliar nutrition have the greatest potential due to their economic and environmentally friendly nature. Therefore, to understand ammonium thiosulphate (ATS)-mediated foliar S nutrition effects on yield, protein content, mineral biofortification, and sulphur economy of rainfed pearl millet under diverse crop establishment systems, a field study was undertaken. The results highlighted that pearl millet grain and protein yield was significantly higher under no-tillage +3 t/ha crop residue mulching (NTCRM) as compared to no-tillage without mulch (NoTill) and conventional tillage (ConvTill), whereas the stover yield under NTCRM and ConvTill remained at par. Likewise, grain and stover yield in foliar S application using ATS 10 mL/L_twice was 19.5% and 13.2% greater over no S application. The sulphur management strategy of foliar-applied ATS 10 mL/L_twice resulted in significant improvement in grain protein content, protein yield, micronutrient fortification, and net returns (₹ 54.6 × 1000) over the control. Overall, ATS-mediated foliar S nutrition can be an alternate pathway to S management in pearl millet for yield enhancement, micronutrient biofortification and grain protein content increase under ConvTill, as well as under the new NTCRM systems.

## 1. Introduction

Pearl millet (*Pennisetum glaucum* L. R. Br. Emend Stuntz) is a kingpin of food and fodder security, as well as the rural economy of rainfed production systems across the globe [1]. Owing to its greater degree of resilience to climatic adversities, it is capable of feeding human and livestock populations under fragile agroecosystems [2]. Furthermore, it is more nutritious than other cereals, has a higher amount of tolerance to saline and acidic soils, and is well adapted to marginal lands with low productivity [3]. Pearl millet is one of the most drought-hardy cereals and can be cultivated under a minimal amount of annual rainfall (300–500 mm), which makes it a default choice for rainfed dryland ecologies [4]. India is the largest producer of pearl millet in the world, as it covers ~30 percent of global area and production of the crop [5]. The country produces 8.61 million tons of pearl millet grains annually, with a productivity of 1243 kg/ha over an area of 6.93 million ha [6]. Interventions including advancement of agronomic practices, such as land configuration, along with standardization of doses of major nutrients and varietal improvement, have led to significant contributions to enhanced productivity over the past several decades [7]. However, the increment in productivity is stagnating and not keeping pace with the burgeoning population. In addition, micronutrient and protein malnutrition, primarily due to quality concerns in staple diets, has surfaced as a mountainous challenge, particularly in emerging economies [8]. Therefore, to tackle the issue of food and nutritional security in developing countries, there is a need to adopt an integrated pathway by judiciously combining the use of productivity boosters and novel production approaches to improve the quality and quantity of major crops of the region [9].

Scanty and erratic precipitation patterns in the arid and semi-arid regions makes pearl millet more prone to moisture stress during critical stages of its ontogeny, leading to significant yield losses [10]. Because moisture stress or drought affects the availability of soil-applied nutrients through its effect on the movement of nutrients via diffusion and mass flow [11], nutrient mineralization [12] and synergistic or antagonistic interactions among nutrients, etc., along with external environmental factors, moisture conservation and nutrient management, are the two most crucial interrelated aspects to realize the full yield potential of dryland crops, including pearl millet [10]. Among diverse conduits to reduce moisture and nutrient stress in rainfed drylands, conservation agriculture (CA) and foliar feeding of nutrients seem to have the greatest potential to tackle the problem in a more economic and environmentally sound manner [2,3,7,10].

CA practices—no-tillage, crop residue mulching, and sustainable rotations—are cost-efficient agrotechniques with low carbon and energy footprints, owing to the savings of fossil fuels from a reduced number of tillage operations, as well as low greenhouse gas emissions associated with energy consumed in manufacture, transport, repair and use of machines, which lowers global warming potential [4]. Sustainable production with environmental safety under long-term CA is well documented [4]. The superiority of zero tillage was reported in rainfed ecologies, with 4 t/ha mustard residues with an increased pearl millet yield of 22.3% as compared to conventional tillage (ConvTill) [7].

Similarly, sulphur, the fourth important nutrient in the plant nutrition aspect of Indian agriculture, and its widespread deficiency in soils have been reported across South Asia [13,14]. As nitrogen is a major nutrient with significant cost for its supply in fertilizers, its efficient use should be ensured. Sulphur, due to its synergistic interaction with nitrogen [15]—a limiting nutrient in Indian soils, especially in drylands—can play a major role in improving the quality and quantity of production by ensuring optimal utilization of applied nitrogen [16] and improving protein content in grains. Based on previous studies, a dose of S 30 kg/ha as a soil application under zero tillage with mulching of 4 t/ha mustard residues increased protein yield, as well as grain yield, of pearl millet [7]. However, the availability and cost of sulphur fertilizers often make them beyond the reach of the poor farming community of drylands.

S deficiency in Indian soils is generally corrected in oilseeds and legumes through S-containing fertilizers as a soil application [17]. The fact that the amount of S recommended for soil application is considerably high not only raises concerns about the cost of fertilizer and its tedious handling during transportation and application but also about significant losses through leaching, etc. This results in low recovery, as well as inefficient use of S. Moreover, under biotic and abiotic stresses, the ability of plants to absorb nutrients through the roots is reduced due to decreased root activity, whereas foliar application addresses all these issues single-handedly. Therefore, is the most effective method and a crosscut for plant nutrition under such adverse scenarios [18]. However, the sources of S for foliar application in rainfed crops and ecologies not been evaluated so far. Therefore, to alleviate S deficiency in a more economic and environmentally sound manner, alternate low-cost, low-volume sources of sulphur suitable for foliar application were tested to optimize its dose and recovery. Furthermore, we carried out field experiments considering the knowledge gap in foliar S management effects on biofortification and quality aspects in CA, as well as in ConvTill.

## 2. Results

### 2.1. Growth Characteristics

The CA system had a significant effect on the number of tillers per meter of row length, dry matter accumulation, leaf area index, and SPAD value at various stages of crop growth (Table 1) over ConvTill. The plant stands, in terms of the number of tillers per meter row length, were found to be densest under ConvTill, followed by NTCRM, at the 30- and 60-day stage; however, density was statistically at par across the treatments. The plant stands differed significantly only at the harvest stage under NTCRM, which was statistically higher compared to ConvTill, as well as NoTill without crop residue mulching. Initially, the partitioning of dry matter to the plant system was slow. A lapse of 60 days after sowing and onward resulted in a peak of growth and development; therefore, plants accumulated the maximum amount of dry matter at 60 days (coincides with flowering and heading) and at the harvest stage. At the harvest stage of the crop, a significantly higher amount of dry matter was accumulated by the plants under NTCRM over NoTill. However, it was on par with ConvTill. Relative leaf area over land area, the leaf area index (LAI) at the 30-days and 60-day stage of crop growth was found to be significantly higher in NTCRM as compared to ConvTill, whereas the canopy cover under NoTill was statistically on par with that under ConvTill. There was a non-significant difference in SPAD value at the 30-day stage; however, at the 60-day stage of crop growth, NTCRM registered the highest value of SPAD, which was significantly higher compared to NoTill without residues and remained at par with ConvTill without residue. However, there was no significant difference in the SPAD values of ConvTill and NoTill without residue.

Likewise, foliar application of ATS at various doses and intervals had a significant effect on growth attributes under the study at various stages of crop growth compared to no sulphur application (Table 1). The plant stand was found to be significantly higher under the soil application of sulphur at 30 kg/ha compared to the control, as well as foliar-applied ATS. Furthermore, the plant stands of the plots sprayed with ATS either at 5 mL/L or 10 mL/L once or twice were statistically on par with RDS at the 30-day stage. However, at the 60-day stage of crop growth, a comparatively denser plant stand was recorded in ATS 10 mL/L_twice, on par with the other treatments, except the control and foliar application of ATS 5 mL/L_once. Contrarily, the plant stand was found to be comparatively denser with RDS and remained at par with foliar-applied ATS 5 mL/L twice and 10 mL/L twice at maturity. Accumulation of dry matter by the plants at the 30-day stage followed a similar pattern. However, a sharp increase in dry matter partitioning was noticed in RDS, as well as foliar-applied ATS 5 mL/L_twice and 10 mL/L_twice compared to the control and foliar-applied ATS 5 mL/L_once at the 60-day stage and at harvest. The treatment, foliar application of ATS 10 mL/L_twice, achieved a similar level of dry matter accumulation as that of RDS at the 60-day stage and at harvest. The magnitude of LAI and SPAD values were also increased with each successive level of foliar-applied sulphur as compared to the control. However, the maximum values of these parameters were observed with foliar application of ATS at 10 mL/L twice, which was significantly higher than the control, but remained on par with soil-applied S at 30 kg/ha and the other foliar-applied S levels at the 30-day and 6-day stages.

### 2.2. Yield Attributes

The yield-attributing characteristics of pearl millet—earhead per meter row length, length of earhead, grain weight per earhead, and 1000-grain—were also found to vary significantly with crop establishment methods (Table 2). The CA practice of NTCRM produced significantly longer earheads with heavier grains, as evidenced by higher grain weight per earhead and test weight. However, the number of earheads per meter row length was found highest under NTCRM, which was on par with ConvTill but significantly higher as compared to NoTill.

The yield-attributing traits of pearl millet were also found to vary significantly with sulphur application methods, dose, and interval. The highest values of these traits were recorded with RDS (S 30 kg/ha), but it was found to be statistically on par with foliar application of ATS 10 mL/L_twice. Although a significant increase in these traits was observed with increasing levels of foliar-applied sulphur through ATS compared to the control, foliar application of ATS 10 mL/L_twice only produced a similar level of these traits as that of RDS (S 30 kg/ha). However, the foliar application of ATS 5 mL/L_twice resulted in statistically on par values of the test weight as obtained with best-performing treatment of soil application and foliar application of ATS 10 mL/L_twice.

### 2.3. Yield and Harvest Index

The grain and stover yield and harvest index of pearl millet (Table 2) showed a significant variation with CA practice. NTCRM resulted in a significantly higher grain and stover yield compared to NoTill, and similar observations were recorded for harvest index. Although the practice of NTCRM and ConvTill produced statistically similar stover yields, the grain yield of the former was found to be significantly higher than that of the latter.

RDS as a soil application to pearl millet enhanced the grain and stover yields by 22.0 and 14.1%, respectively, over the control. The grain and stover yields in foliar-applied ATS 10 mL/L_twice were statistically on par with those of RDS, with a yield increase of 19.5% and 13.2%, respectively, which is comparable to the yield increase in RDS. The grain and stover yield levels under foliar-applied ATS 10 mL/L_once and ATS 5 mL/L_twice were on par but significantly higher than those under foliar application of ATS 5 mL/L_once.

### 2.4. Protein Content and Protein Yield

Crop establishment systems and sulphur management strategies both affected the protein content and grain protein yield of pearl millet compared to the control (Figure 1). Although the grain protein content across the crop establishment systems was not changed significantly, numerically, the highest amount of grain protein content was recorded in NTCRM. The highest grain protein and protein yield were recorded under RDS and increased with increasing levels of foliar ATS across the establishment systems.

NTCRM, NoTill and ConvTill denote crop establishment systems (no-tillage with crop residue mulching at 3 t/ha, no-tillage and conventional tillage, respectively), and RDS, ATS 5 mL/L_once, ATS 10 mL/L_once, ATS 5 mL/L_twice and ATS 10 mL/L_twice indicate recommended dose of S at 30 kg/ha, one foliar application of ammonium thiosulphate (ATS) at 5 mL/L and 10 mL/L, respectively, at the 4–6-leaf stage and two foliar sprays of ATS at 5 mL/L and 10 mL/L, respectively, at the 4–6 leaf stage 30 days after the first application. The bar represents the least significant difference (CD_0.05_).

### 2.5. Economics

The cost of cultivation of pearl millet crop varies across establishment methods. Cost was lowest under NoTill and highest under ConvTill (Table 2). The practice of NTCRM achieved maximum net returns, which were significantly higher compared to those under ConvTill and NoTill. However, the net returns from NoTill were significantly lower than those under NTCRM, but the highest B:C ratio was recorded under the former, followed by the latter. RDS (30 kg/ha) and foliar application of ATS at 10 mL/L_twice and 5 mL/L_twice remained on par with each other and achieved significantly higher net returns and B:C ratios.

### 2.6. Micronutrient Content vs. Grain Yield

The micronutrient content in grain significantly influenced the crop establishment systems, as well as S management strategies (Figure 2). In general, the highest contents of Fe, Zn, Mn and Cu were recorded under NTCRM, with significantly higher concentrations than under the other establishment systems. Grain Zn and Mn contents improved with the successive levels of S, either through RDS or through the foliar spray of ATS, whereas a reverse trend was observed in the content of Fe and Cu. The antagonistic effect of RDS on the content of Fe and Cu was more pronounced as compared to foliar application.

NTCRM, NoTill and ConvTill denote crop establishment systems (no tillage with crop residue mulching at 3 t/ha, no tillage and conventional tillage, respectively), and RDS, ATS 5 mL/L_once, ATS 10 mL/L_once, ATS 5 mL/L_twice and ATS 10 mL/L_twice indicate recommended dose of S at 30 kg/ha, one foliar application of ammonium thiosulphate (ATS) at 5 mL/L and 10 mL/L, respectively, at the 4–6-leaf stage and two foliar sprays of ATS at 5 mL/L and 10 mL/L, respectively, at the 4–6-leaf stage 30 days after the first application. The bar represents least significant difference (CD_0.05_).

### 2.7. Sulphur Economy

It is evident from the Table 3 that the total uptake of S and residual S is inversely related, as the increase in total S uptake by pearl millet in S-supplied plots resulted in less residual S left in the soil and vice-versa in the control. In general, the apparent S balance was computed to be negative in soil across crop establishment methods and S management treatments. Although the S balance was negative in all crop establishment methods, the balance was more negative under ConvTill. The results indicated that the practice of retaining crop residues resulted in a higher supply of S and, consequently, a less negative S balance in the soil as compared to no crop residue under both tillage systems. S losses were also higher in ConvTill, whereas the soil either under mulching (NTCRM) or without disturbance (NoTill) replenished the S losses either by mineralization of S present in retained crop residue or by slow oxidation of sulphur pools inherently present in the soil, respectively.

The method of S fertilization also influenced apparent S balance in the soil. A similar pattern of inverse relationship in total S uptake and residual S was also observed with successive levels of S compared to control. A positive S balance was recorded only in RDS (S 30 kg/ha). However, the S balance was less negative with successive levels of S through the foliar spray of ATS and was highly negative in control, more negative in a foliar spray of ATS 5 mL/L_once and less negative in a foliar spray of ATS at 10 mL/L_twice. A negative S balance with no S application to a dose of S at 15 kg/ha and a positive S balance only with 30 kg and 45 kg S/ha was recorded in pearl millet in a two-year study [19].

## 3. Discussion

The improvement in growth characteristics and yield of pearl millet is mainly attributed to the use of conserved moisture in residue-covered plots [20] and the continuous supply of soil moisture to the crops. Adequate availability of moisture to plants resulted in cell turgidity and, eventually, high meristematic activity, leading to more foliage. Elemental S, an immobile form of S in the soil, needs to be oxidized into the plant-available form of sulphate (SO_4_^2−^) before uptake by crop roots or microbes [21]. The oxidation of elemental S is governed by microbial activity, which is strongly affected by soil moisture content and soil temperature [22]. Moreover, a plant nutrient has to travel a long path through the plant system (phloem) after its absorption and translocation through the plant roots to reach the leaves and grains, whereas in the foliar application, the nutrients needed by the plant rapidly enter the phloem and reach the target sites [18].

The biosynthesis of essential amino acids (cystine, cysteine and methionine), the building blocks of plant protein, largely depends on S availability, as well as its fertilization in soil; hence, not only RDS but also successive levels of ATS significantly increased the protein content and protein yield. Moreover, the well-known synergistic interaction between S and N [23] resulted in higher availability of N, which was ultimately reflected in higher grain protein content. Moreover, there is a strong correlation between uptake and assimilation of S and N in plants [15], as S at its optimum dose not only helps crops to reach their full yield potential but also improves quality in terms of protein content and N use efficiency [16]. Because protein yield is a product of protein content in grains and grain yield, it was recorded highest under NTCRM with RDS, followed by ATS 10 mL/L_twice.

The low cost of cultivation in NoTill treatments was mainly due to a reduction in the cost of land preparation and manual weed control. Higher net returns were recorded under NoTill due to more returns from proportionately higher yield as compared to the cost involved under this crop establishment system. Lower B:C ratio under NTCRM in comparison to NoTill was mainly attributed due to lower returns from residue in comparison to the cost involved. Similar findings were also reported by Gupta et al. [24], Choudhary et al. [7] and Ruxanabi et al. [25].

The micronutrients in grain were significantly higher under NTCRM due to the favorable microclimatic effect of crop residue mulching and release of these nutrients after the decomposition of the organic residues in soil, which replenished the micronutrient pool in real time [4]. Whereas the higher Zn and Mn contents in grains with successive levels of S are due to the synergistic effect of S application on Mn uptake and the absence of competition among ions on the site of absorbance. Likewise, the uptake of Fe and Cu [26] remained lower owing to significantly less Fe and Cu content in the grains (Figure 2).

In general, ConvTill and NoTill without residues were statistically similar with respect to negative apparent S balance, owing to one-sided S losses in disturbed and exposed soil. The maximum (less negative) apparent S balance was recorded with chickpea residue mulching at 3 t/ha under NTCRM (Table 3). The improvement in S content under NTCRM could be ascribed to favorable moisture conditions in the soil maintained for a relatively long period and improvement in the available S status of soil through decomposition of crop residues. Thus, the favorable moisture condition and improved nutritional environment led to higher translocation and assimilation of nutrients to grains and stover [27,28]. Furthermore, the application of crop residue moderates soil pH through the respiratory CO_2_ of microbes and the formation of organic acids during decomposition. Moreover, decomposition products might give rise to natural complexing agents that solubilize the nutrients already present in soil [10,29,30,31]. Because the uptake of the nutrient is a function of nutrient content and biomass production, the significant increase in S content coupled with increased yield under NTCRM enhanced the total uptake of S [22].

The apparent S balance in pearl millet was only positive with respect to RDS at 30 kg/ha, whereas it was negative throughout in the case of either level of foliar S feeding under ATS. However, the balance negativity was successively reduced as the dose of S increased through ATS but remained negative, owing to accelerated S uptake by pearl millet as catalyzed by combined S and N supply.

## 4. Materials and Methods

### 4.1. Experimental Site

The field trials were conducted on sandy loam soils from 2020 to 2021 at the research farm of the Division of Agronomy, Indian Agricultural Research Institute, New Delhi. Geographically, the site lies at a latitude of 28°40′ N and a longitude of 77°12′ E with an altitude of 228.6 m above the mean sea level.

### 4.2. Soil and Climate

This location has a typical semi-arid and sub-tropical climate characterized by hot, dry summers and cool winters. The mean annual rainfall is 650 mm, and more than 80% generally occurs during the southwest monsoon season (July to September), with mean annual evaporation of 850 mm. The total rainfall received during the crop-growing period from July to October was 615.3 mm in 2020 and 1484.2 mm in 2021 (Figure 3).

The present experiment was conducted on a fixed layout consisting of permanent plots of no-tillage with crop residue mulching (NTCRM), no-tillage without mulching (NoTill) and conventional tillage (ConvTill) in *Typic haplusteps* soil. In general, the soil of all the three tillage systems under study was found to be low in alkaline permanganate-oxidizable nitrogen, medium in available P and low in 1 N ammonium acetate-exchangeable potassium K. The critical level of CaCl_2_-extractable sulphur for pearl millet grown on alluvial soils in the Trans-Gangetic Plains of north India varies from 9.0 to 10.0 mg/kg soil [32]; thus, the experimental soil was inherently low in available S, and the response of pearl millet to S application was expected in the experimental field. (Table 4).

### 4.3. Treatment Details

The experiment was designed with treatment combinations of three crop establishment practices comprising no-tillage with residue (NTCRM) and without crop residue, mulching (NoTill) and conventional tillage (ConvTill) in main plots and six strategies of S management, namely control, soil application of 30 kg/ha S (RDS) through elemental S, one foliar application of ammonium thiosulphate (ATS) at 5 mL/L and 10 mL/L, respectively, at the 4–6-leaf stage and two foliar sprays of ATS at 5 mL/L and 10 mL/L, respectively, at the 4–6-leaf stage and 30 days after the first application as sub-plots.

### 4.4. Crop Establishment and Treatment Application

The conventional field was prepared with a deep ploughing by a disc plough, followed by two passes of a disc harrow. The field was planked in the last tillage to create a uniform seedbed of fine tilth. The no-tillage plots were prepared by knocking down all weeds with a pre-sowing spray of glyphosate 41% SL at 0.1%. A composite variety of pearl millet, namely ‘Pusa-443’, was selected for experimentation. The crop was sown on 10 July in 2020 and 16 July in 2021 using a seed rate of 4 kg/ha at a uniform row spacing of 45 cm. The previous season’s chickpea residues at 3 t/ha (0.15% S) were applied to a pre-designated NTCRM plot just after the sowing, maintaining a uniform layer over the soil for effective moisture conservation. Plant rectangularity was maintained by thinning at the time of intercultural operations at 15 DAS, keeping an intra-row distance of 15 cm.

All the main plots were fertilized with a uniform recommended dose of N:P_2_O_5_:K_2_O at 60:40:30 kg/ha through urea, single super phosphate and muriate of potash, respectively. The precalculated amount of nitrogen was applied in two split doses; the first half dose was basally applied, and the second half was top-dressed at the pre-flowering stage to maximize the N-use efficiency. The basal amount of N, along with P_2_O_5_ and K_2_O, were mixed at the time of final land preparation in the ConvTill plot, whereas it was applied in a subsurface band about 5 cm to the side of pearl millet seeds during sowing in NTCRM and NoTill plots. The amount of nitrogen supplied through ATS was deducted at 14 per cent (350 g, 700 g, 700 g and 1400 g/ha for respective foliar sprays) from the dose of nitrogen to be supplied through urea in treated plots. The recommended dose of sulphur (RDS at 30 kg/ha), as per the treatments through elemental sulphur (90% S), was side-dressed just after sowing of pearl millet. The amount of sulphur supplied through ATS was computed at 21.5% S. Therefore, amounts of 537.5 g, 1075 g, 1075 g and 2150 g/ha were added to the field through ATS foliar sprays at 5 mL/L once, 10 mL/L once, 5 mL/L twice and 10 mL/L twice, respectively. Weeds were managed by a pre-emergence spray of Atrazine at 0.5 kg a.i./ha.

### 4.5. Soil Physicochemical Properties Analysis

The soil samples were collected from furrow slices (0–15 cm soil depth), air-dried and were passed through a 2 mm sieve. Soil pH and electrical conductivity (EC) were analyzed using air-dried soil in a ratio of 1:2.5 of soil weight and double-distilled water [33]. A compact pH meter (Systronic; pH System 361) and a compact electric conductivity meter (Systronic; Conductivity TDS Meter 308) were used. Available N in the soil profile was estimated after alkaline 0.32% KMNO_4_ solution in a ratio of 1:10, heated gradually and liberated ammonia was absorbed in 2% boric acid solution and titrated against 0.02 N H_2_SO_4_ [34]. Available P was extracted from the soil with 0.5M NaHCO_3_ at a constant pH of 8.5 [35]. Available K was extracted with 1 N NH_4_OAc in a 1:5 ratio, shaken for 5 min and filtered through Whatman No. 1 filter paper [36]. The available S content was extracted using 0.15% CaCl_2_ solution in a 1:5 ratio by turbidimetric method [37] at the beginning of the experiment to estimate the initial status, as well as after the crop harvest to estimate residual S in the soil.

### 4.6. Plant Sampling, Data Collection and Plant Chemical Analysis

All the growth parameters and yield attributes were recorded following standard procedure [38]. The number of tillers was recorded at 30 DAS, 60 DAS and at harvest from pre-designated observation. Rows were recorded using a meter scale and by counting the tillers confined in 1 m row length. Dry matter was recorded by sampling out five representative plants at 30 DAS, 60 DAS and at harvest maturity. The plant samples were oven-dried for a constant dry weight at 65 °C in a hot-air oven, and the average weight per tiller was recorded by standard electronic digital balance.

Plants with uppermost fully expanded leaves were selected, and the leaf chlorophyll concentration was observed midway between the stalk and the tip of the leaf with a portable chlorophyll meter (SPAD-502 Minolta, Tokyo, Japan). The recorded values are expressed in arbitrary absorbance (or SPAD) values. In addition, leaf area was measured with a leaf area meter (LICOR-3000, Linclon, OR, USA).

Yield attributes such as earheads per meter row length were recorded by counting the effective tillers in 1 m length. The length of five representative earheads was recorded with a standard meter scale and expressed in cm. Five earheads were randomly selected from the observation plot of each replication; threshed and grain weight per earhead were recorded separately at 12% moisture level with an electronic digital balance and expressed in grams. Likewise, 1000-grain weight was recorded from the same earheads. The plants from the net plot were harvested from the ground level and were left for sun drying in situ. The whole plant of pearl millet was threshed with a Pullman thresher. Grains were winnowed, air-dried and weighed to express treatment-wise yield in t/ha at a 12% moisture level. The weight of the stalk was recorded separately and used to estimate stover yield at a 15% moisture level.

The grain and stover samples from the observation area were drawn from respective treatment at the time of harvest. The samples were prepared by drying in a hot-air oven at 60 °C for 48 h, ground into fine powder in a ‘Macro-Wiley’ mill and passed through 40 mesh sieves. From each replication, 0.5 g of powdered grain and stover samples was taken and analyzed for total S content, employing the turbidimetric method after wet digestion with diacid mixture (HNO_3_:HClO_4_) [37] and expressed in %. The uptake of sulphur by grain and stover was computed using the following expressions:$$S\;\mathrm{uptake}\;\left(\mathrm{kg}/\mathrm{ha}\right)=\frac{S\;\mathrm{content}\;\left(\%\right)\;\mathrm{in}\;\mathrm{sample}\;\times \;\mathrm{Dry}\;\mathrm{matter}\;\mathrm{on}\;\mathrm{oven}\;\mathrm{dry}\;\mathrm{weight}\;\mathrm{basis}\;\left(\mathrm{kg}/\mathrm{ha}\right)}{100}$$
Total uptake of S (kg/ha) = S uptake by grain (kg/ha) + S uptake by stover (kg/ha)

The micronutrients zinc (Zn), iron (Fe), copper (Cu) and manganese (Mn) (DTPA extractable) in pearl millet grain were estimated from the same diacid digested samples by and ‘Element AS-AAS4141’ (ECI Ltd.) double-beam atomic absorption spectrophotometer as per the Lindsay and Norvell method [39].

### 4.7. Protein Content and Yield Calculation

The total nitrogen content of grain estimated with the modified Kjeldahl method [33] was multiplied with a standard factor of 6.25 as suggested by (Crompton and Harvois, 1969) [40]. The grain protein yield was worked out as per the following expression:$$\mathrm{Grain}\;\mathrm{protein}\;\mathrm{yield}\;\left(\mathrm{kg}/\mathrm{ha}\right)=\frac{\mathrm{Protein}\;\mathrm{content}\;\left(\%\right)\;\mathrm{in}\;\mathrm{grains}\;\times \;\mathrm{Grain}\;\mathrm{yield}\;\left(\mathrm{kg}/\mathrm{ha}\right)}{100}$$

### 4.8. Apparent S Balance

Apparent S balance was estimated after each harvest during both years. The S balances were determined by differences between the inputs in the form of added S (either through elemental sulphur or ATS) and crop residue and outputs in the form of total uptake by pearl millet [41].

### 4.9. Economics

Economics of different crop establishment methods and sulphur levels was worked out by taking into account the unit cost of inputs at prevailing prices and output (grain and stover yield) minimum support price (MSP) of the respective year. The net returns benefit: cost (B:C) ratio was determined using following expressions for each treatment:Gross returns (₹/ha) = Grain yield MSP (₹/ha) + Stover yield Sale price (₹/ha)
Net returns (₹/ha) = Gross returns—cost of cultivation
$$\mathrm{Benefit}:\;\mathrm{cost}\;\mathrm{ratio}\;=\frac{\mathrm{Net}\;\mathrm{returns}\;\left(₹/\mathrm{ha}\right)}{\mathrm{Cost}\;\mathrm{of}\;\mathrm{cultivation}\;\left(₹/\mathrm{ha}\right)}\times 100$$

### 4.10. Statistics

Analysis was carried out in a split-plot design with three replications. The growth and yield data were subjected to the technique of analysis of variance (ANOVA) for statistical analysis in a split-plot design. The CD values were calculated to compare the treatment means, and results are reported at a 5% significance level [42].

## 5. Conclusions

The performance of pearl millet in all the crop establishment methods, as well as sulphur application methods, was comparable, each with their own pros and cons. The practice of no-tillage, along with crop residue mulching (NTCRM), produced significantly greater yield with superior quality grains. Feeding of sulphur at the early growth stage of 4 to 6 leaves, as well as 30 days after the first application, was critical for the improvement in growth, yield and quality of grains, as evidenced by the significant yield and protein content difference between single and double sprays of ATS at varying doses. However, the apparent S balance was more negative when a foliar spray of ATS was applied without conservation agriculture practices, whereas it was found to be more environmentally sound under conservation agriculture management system, as it significantly replenished the S outflow either through crop uptake or other inevitable losses. Hence, foliar feeding of 10 mL/L ATS twice at the early growth stage and flowering stage in pearl millet under the crop establishment method of no-tillage, along with crop residue mulching at 3 t/ha, is recommended in rainfed ecologies.

## Figures and Tables

**Figure 1 plants-11-00943-f001:**
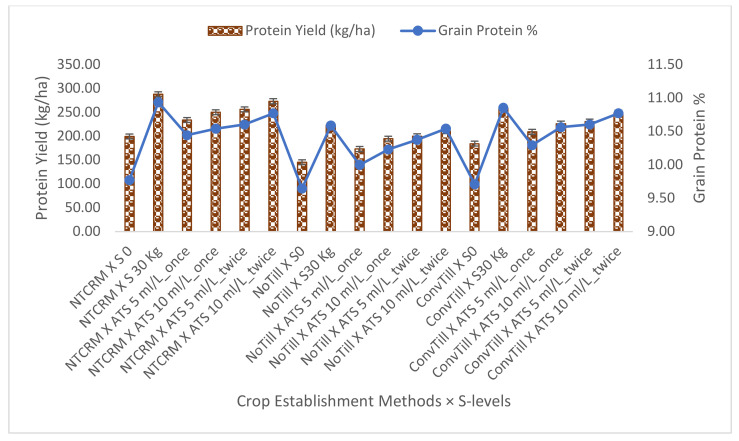
Grain protein content (%) and grain protein yield (kg/ha) of pearl millet as influenced by crop establishment methods and sulphur management strategies (mean of two years of data).

**Figure 2 plants-11-00943-f002:**
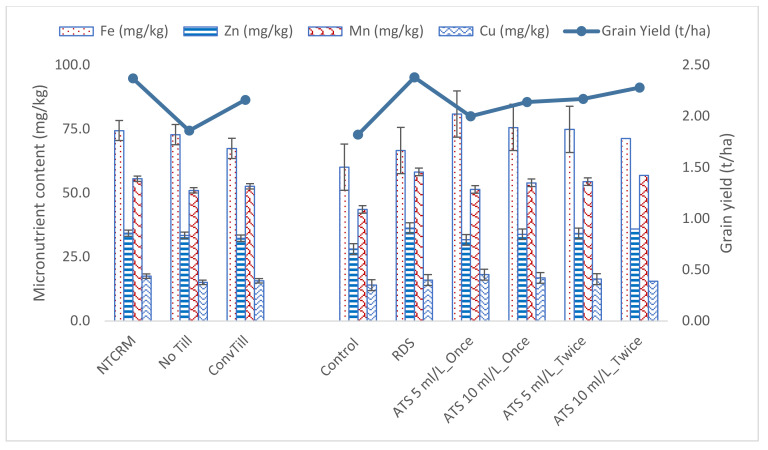
Micronutrient content (mg/kg) in grain and grain yield (t/ha) of pearl millet as influenced by crop establishment methods and sulphur management strategies (mean of two years of data).

**Figure 3 plants-11-00943-f003:**
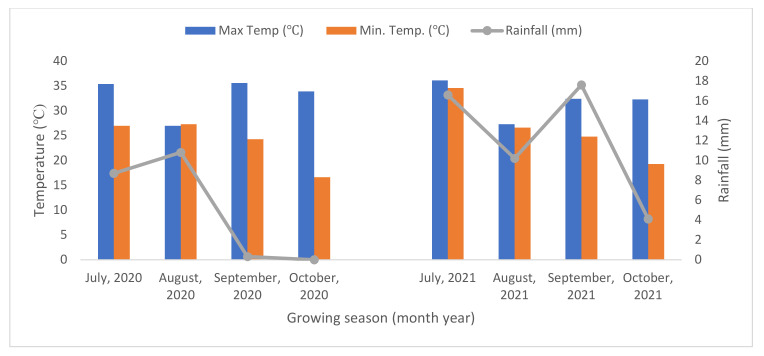
Mean monthly temperatures and rainfall received during pearl millet growing season at Research Farm IARI field trial site.

**Table 1 plants-11-00943-t001:** Effect of crop establishment system and sulphur nutrition levels on growth parameters of pearl millet (mean of two years of data).

Treatments	No. of Tillers/m Row			Dry Matter Production (g/Tiller)			Leaf Area Index		SPAD Value	
	30 DAS	60 DAS	At Maturity	30 DAS	60 DAS	At Maturity	30 DAS	60 DAS	30 DAS	60 DAS
** *Crop establishment systems* **										
NTCRM	9.72	19.33	23.67	11.15	56.65	101.43	1.17	4.66	43.51	51.84
NoTill	9.28	17.72	20.94	10.82	50.37	96.56	1.07	4.07	41.63	48.47
ConvTill	9.61	19.78	21.89	10.92	53.59	100.06	1.10	4.24	44.22	49.93
SEm±	0.28	0.73	0.31	0.07	0.71	0.92	0.02	0.08	0.02	0.56
CD (*p = 0.05*)	NS	NS	1.27	NS	2.84	3.70	0.06	0.32	N/S	2.28
** *S management strategies* **										
Control	6.22	16.11	19.22	9.64	42.66	81.00	0.99	3.95	40.81	46.28
RDS	10.22	20.22	24.33	11.57	61.38	113.16	1.13	4.47	45.06	51.97
ATS 5 mL/L_Once	10.11	18.44	21.00	10.56	49.10	95.03	1.09	4.15	42.32	47.49
ATS 10 mL/L_Once	9.89	18.89	22.11	11.81	51.62	98.90	1.16	4.30	43.93	50.48
ATS 5 mL/L_Twice	10.22	19.67	22.67	10.49	55.93	101.02	1.13	4.44	42.27	51.48
ATS 10 mL/L_Twice	10.56	20.33	23.67	11.72	60.53	106.99	1.17	4.63	44.31	52.79
SEm±	0.38	0.64	0.68	0.28	1.19	2.18	0.03	0.13	0.85	0.87
CD (*p = 0.05*)	1.09	1.86	1.97	0.81	3.46	6.33	0.08	0.37	2.45	2.52

NTCRM, NoTill and ConvTill denote crop establishment systems (no tillage with crop residue mulching at 3 t/ha, no tillage and conventional tillage, respectively), and RDS, ATS 5 mL/L_once, ATS 10 mL/L_once, ATS 5 mL/L_twice and ATS 10 mL/L_twice indicate recommended dose of S at 30 kg/ha, one foliar application of ammonium thiosulphate (ATS) at 5 mL/L and 10 mL/L, respectively at the 4–6-leaf stage and two foliar sprays of ATS at 5 mL/L and 10 mL/L, respectively, at the 4–6-leaf stage and 30 days after the first application.

**Table 2 plants-11-00943-t002:** Effect of crop establishment system and sulphur nutrition levels on yield attributes, yield and economics of pearl millet (mean of 2 years data).

Treatment	Earheads/m Row	Earheads Length (cm)	Grain Weight/Earhead (g)	Test Weight (g)	Grain Yield (t/ha)	Stover Yield (t/ha)	Harvest Index (%)	Cost of Cultivation (×10^3^ ₹/ha)	Net Return (×10^3^ ₹/ha)	Benefit:Cost Ratio
** *Crop establishment systems* **										
NTCRM	16.06	29.33	19.50	8.48	2.37	7.66	23.61	25.18	56.50	2.24
NoTill	14.17	27.31	18.70	8.18	1.86	6.78	21.51	20.43	46.71	2.28
ConvTill	15.50	28.61	18.86	8.27	2.16	7.44	22.47	27.04	49.10	1.81
SEm±	0.37	0.14	0.10	0.05	0.01	0.04	0.17	-	0.34	0.02
CD (*p = 0.05*)	1.48	0.57	0.42	0.19	0.05	0.16	0.63	-	1.36	0.07
** *S management strategies* **										
Control	13.11	26.76	16.39	7.65	1.82	6.50	21.74	22.85	42.21	1.86
RDS	17.11	29.81	20.42	8.62	2.38	7.83	23.24	25.60	56.82	2.24
ATS 5 mL/L_Once	14.78	27.51	18.57	8.23	2.00	7.00	22.16	23.57	47.45	2.03
ATS 10 mL/L_Once	15.00	28.14	19.47	8.36	2.14	7.36	22.49	23.95	51.47	2.17
ATS 5 mL/L_Twice	15.33	28.53	19.53	8.43	2.17	7.42	22.65	24.30	52.07	2.16
ATS 10 mL/L_Twice	16.11	29.74	19.73	8.56	2.28	7.66	22.90	25.05	54.62	2.20
SEm±	0.40	0.56	0.25	0.08	0.04	0.09	0.24	-	0.99	0.04
CD (*p = 0.05*)	1.56	1.67	0.73	0.24	0.10	0.26	0.69	-	2.88	0.12

₹ = Indian rupee. NTCRM, NoTill and ConvTill denote crop establishment systems (no tillage with crop residue mulching at 3 t/ha, no tillage and conventional tillage, respectively), and RDS, ATS 5 mL/L_once, ATS 10 mL/L_once, ATS 5 mL/L_twice and ATS 10 mL/L_twice indicate recommended dose of S at 30 kg/ha, one foliar application of ammonium thiosulphate (ATS) at 5 mL/L and 10 mL/L, respectively, at the 4–6-leaf stage and two foliar sprays of ATS at 5 mL/L and 10 mL/L, respectively, at 4–6-leaf stage and 30 days after first application.

**Table 3 plants-11-00943-t003:** Sulphur economy (mean of two years of data).

Treatment	Available Sulphur (kg/ha) (i)	Sulphur Applied (kg/ha) (ii)	Total Available Sulphur (kg/ha) (A = i + ii)	S Uptake by Grains (kg/ha) (iii)	S uptake by Stover (kg/ha) (iv)	Total S Uptake (kg/ha) (a= iii + iv)	Residual Sulphur (kg/ha) (b)	Apparent Balance (kg/ha) {A-(a + b)}
** *Crop establishment systems* **								
NTCRM	19.2	10.31 *	24.76	6.85	11.69	18.54	17.44	−6.47
NoTill	18.3	5.81	23.86	4.88	9.43	14.31	19.68	−9.89
ConvTill	17.8	5.81	23.36	6.01	10.80	16.81	17.84	−11.05
SEm±	-	-	-	0.03	0.13	0.15	0.28	0.24
CD (*p = 0.05*)	-	-	-	0.11	0.53	0.59	1.13	0.95
** *S management strategies* **								
Control	18.43	1.50 *	18.43	3.89	8.18	12.07	17.77	−9.90
RDS	18.43	31.5	48.43	7.50	13.03	20.52	21.41	8.01
ATS 5 mL/L_Once	18.43	2.04	18.78	4.92	9.30	14.22	20.14	−13.90
ATS 10 mL/L_Once	18.43	2.58	19.18	5.85	10.37	16.21	18.07	−13.27
ATS 5 mL/L_Twice	18.43	2.58	19.18	6.24	10.71	16.95	16.99	−12.95
ATS 10 mL/L_Twice	18.43	3.65	19.93	7.07	12.26	19.34	15.56	−12.82
SEm±	-	-	-	0.15	0.26	0.33	0.40	0.53
CD (*p = 0.05*)	-	-	-	0.44	0.77	0.96	1.16	1.54

* Sulphur supplied by chickpea residues retained at 3 t/ha was 4.5 kg (S content in residues, 0.15%). NTCRM, NoTill and ConvTill denote crop establishment systems (no tillage with crop residue mulching at 3 t/ha, no tillage and conventional tillage, respectively), and RDS, ATS 5 mL/L_once, ATS 10 mL/L_once, ATS 5 mL/L_twice and ATS 10 mL/L_twice indicate recommended dose of S at 30 kg/ha, one foliar application of ammonium thiosulphate (ATS) at 5 mL/L and 10 mL/L, respectively, at the 4–6-leaf stage and two foliar sprays of ATS at 5 mL/L and 10 mL/L, respectively, at the 4–6-leaf stage and 30 days after the first application.

**Table 4 plants-11-00943-t004:** Soil physicochemical properties of 0–15 cm depth at the start of the trial.

Tillage Plot	Soil *p*H	Soil EC (dS/m)	Bulk Density (g/cc)	Moisture Content (%)		Available N (kg/ha)	Available P (kg/ha)	Available K (kg/ha)	Available S (kg/ha)
				(FC)	(PWP)				
NTCRM	7.1	0.24	1.49	19.38	6.83	234.7	15.0	182.4	19.2
NoTill	7.2	0.25	1.51	17.86	6.46	153.5	14.3	171.6	17.5
ConvTill	7.6	0.32	1.47	18.21	6.69	165.3	14.8	175.6	17.1

## Data Availability

Not applicable.

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
