# Peer review of "No-Tillage with Residue Retention and Foliar Sulphur Nutrition Enhances Productivity, Mineral Biofortification and Crude Protein in Rainfed Pearl Millet under Typic Haplustepts: Elucidating the Responses Imposed on an Eight-Year Long-Term Experiment"

_plants, 2022, doi:10.3390/plants11070943_

Round 1

Reviewer 1 Report

This paper considers an example and a very important staple food crop specially for fragile agro-ecosystems – the pearl millet. The results showed a significant improvement in grain protein content, protein yield, micronutrient fortification and net returns comparing to control. This study showed that foliar feeding of 10 ml/L ATS (ammonium thiodulphate) twice at early growth stage and flowering stage in this crop under the crop establishment method of no-tillage along with crop residue mulching at 3t/ha is recommended under rainfed production systems.

Concerning to the quality of the paper: The title of this paper clearly reflects its content. The objectives are clear and appropriate in the view of this subject matter. The results and discussion of this study has a sound English. In general, references are recent and adequate. So, it is a very interesting study, and I recommend its publication in the present form.

Reviewer 2 Report

The publication is interesting because it concerns low cost agrotechniques with low carbon footprints. Unfortunately, the manuscript present results obtained during one cultivation cycle. For this reason, in publication must be clearly stated that it is a case study. It is not certain whether similar results would be obtained in the next years with such difficult climatic conditions. The rainfall (total rainfall in 2020 was 615.3 mm, in 2021 was 1484.2 mm) may be a factor that strongly modifies the results of the experiment.  In addition, the soil chemical properties at the beginning of the experiment are not aligned - there are too large differences in soil available nitrogen (this problem is explained below).

In my opinion research should continue.

Remarks

Fig. 3.

Mean monthly temperatures and rainfall (Fig. 3) are difficult to analyze. I suggest you present on one graph, each year separately.

Table 1 and Table 2

According to the layout style, tables and figures should appear  after they are mentioned in the text for the first time, so you have to move the Table 1 and 2.

Methodology

Soil Physico-Chemical Properties of 0-15 cm depth at the start of the trial – Table 4:

NTCRM  treatment -  Available N: 234,7 kg/ha

NoTill      treatment   -  Available N: 153,5 kg/ha

ConvTill   treatment  -  Available N: 165,3 kg/ha

The result of the experiment was influenced by the content of available nitrogen in the soil at the start of the trial. Nitrogen is the most important yield-generating element. When the initial conditions of the experiment are so different, the obtained results may be due to nitrogen treatment, not treatment application or sulphur.

Author Response

Incorporated all the suggestions as advised by the esteemed reviwer

Reviewer 3 Report

Dear Authors, you should address my comments highlighted throughout the text 

Author Response

Thank you so much. Since no action/reply is needed, we express our sincere thanks to esteemed reviewer. 

Round 2

Reviewer 2 Report

Paper has been greatly improved. I still maintain that  the rainfall (total rainfall in 2020 was 615.3 mm, in 2021 was 1484.2 mm) and the very diverse content of available nitrogen in the soil at the start of the trial may be the factors that strongly modifies results of the experiment.

To accept this publication, it is necessary to add in the title "The case of study" (No-tillage with residue retention and foliar Sulfur nutrition enhances productivity, mineral biofortification and crude protein in rainfed pearl millet under Typic haplustepts. The case of study ). Without this information, the paper should not be published.

Author Response

Respected Sir/Madam,
